# The Ethnobiology of Contemporary British Foragers: Foods They Teach, Their Sources of Inspiration and Impact

Łukasz Łuczaj [1,*], Monica Wilde [2] and Leanne Townsend [3]

1    Institute of Biology and Biotechnology, University of Rzeszów, Pigonia 1, 35-310 Rzeszów, Poland
2    Napiers the Herbalists, 62 George St, Bathgate EH48 1PD, Scotland, UK; monica@napiers.net
3    James Hutton Institute, Craigiebuckler, Aberdeen AB15 8QH, Scotland, UK; leanne.townsend@hutton.ac.uk
*    Correspondence: lukasz.luczaj@interia.pl; Tel.: +48-60-249-7483

**Abstract:** Foraging in the British Isles is an increasingly popular activity for both personal consumption and for commercial purposes. While legislation and guidelines exist regulating the sustainable collection of wild edibles, the founding principles of the British foraging movement are not well documented. For this research, 36 of the most active foraging instructors of the Association of Foragers were interviewed to understand their background, species collected, sources of knowledge, and problems faced during collection. Altogether, 102 species of leafy vegetables, fruits, fungi, and seaweeds were mentioned as frequently used, while 34 species of roadkill animals were listed, mostly for personal consumption. Instructors reported learning from wild food guidebooks, other foragers, or personal experience. Frequent contact among foragers has led to the standardisation of knowledge and practices among them forming a "new tradition", partly based on old British traditions but modified by influences from other countries and cultures, both in terms of choice of species and processing techniques. Contrary to expectations, foragers rarely reported clashes with nature conservation or forestry managers. The authors argue that knowledge and practice developed by the Association of Foragers (AoF) are sustainable and could be integrated into the British food and nature conservation system.

**Keywords:** foraging; wild food; ecosystem services; food security; roadkill; wild edible plants and mushrooms; edible seaweed; edible fungi

## 1. Introduction

Since the Neolithic revolution over 12,000 years ago, hunting and gathering societies have gradually turned into agriculturalists all over the world [1,2]. It is now rare to encounter human societies that subsist on hunting and gathering, and they are mainly in the tropics or the far north [1]. However, historically human foraging of wild foods has been practiced by nearly every rural society in the world, e.g., in times of famine or war [3]. Producing high-calorie crops to feed large populations using modern farming techniques is efficient but it involves high consumption of fossil fuels and uses pesticides and herbicides that are harmful to the environment [4,5]. Thus, the attention of a growing, ecologically aware public movement, particularly young food activists, is turning towards alternative food systems, such as organic farming and wild food alternatives [4–19]. This has contributed to the growth of foraging teaching and new influences in its practice [14,17].

Although the caloric contribution of wild foods to local diets is quite low compared to staple cultivated foods, these species contribute to diet diversification [10]. These forgotten food resources have, in fact, been shown to contain equally, if not higher amounts, of nutrients than the main commercial crops [11–13].

Agriculture in the British Isles started about 8000 years ago [20], initiating a rapid change from foraging to farming. However, some wild foods—i.e., nettle (*Urtica dioica* L.), fat hen (*Chenopodium* spp.), sorrel (*Rumex* spp.), fruits, and nuts—continued to be consumed alongside farmed grains and livestock during the Middle Neolithic [21,22] and actually

until recent pre-war times [23–26]. The practice of using foraged foods is also documented in the cookbooks written during the Middle Ages [27,28]. However, cookery books at this time typically reflected the upper echelons of society whose preferences were often for imported, rare, or exotic species, the acquisition of which reflected high social status [29]. Commonly available "peasant fare" was rarely recorded until the printing press, available from the 15th century, popularised books, making them cheaper to produce and more widely available [30,31]. Cookery book records between the Middle Ages and the 1930s occasionally mention such dishes as dandelion leaf salad [32,33] but with such rarity that it could be presumed that the use of foraged foods had almost died out due to demographic pressure, as suggested by Albala [34], and during the mass transit to cities during the Industrial Revolution.

The United Kingdom formed the first industrial society in the world and was already highly urbanised by the 19th century. Its local, indigenous foraging traditions probably suffered more loss than in other European countries [26]. People actively involved in foraging declined to about 1% of the population by 1800 [35] but the practice was far from extinct. Books recording foraging practices in the early part of the 20th century demonstrate that the use of wild foods had not died out completely. The inhabitants of the British Isles are usually pictured as a prime example of mycophobia; however, the collection of edible wild mushrooms was locally practiced, even in the Victorian times [36–38]. Cooke's first book on the identification and preparation of edible fungi was reprinted six times from 1862 to 1898, and another on fungi reprinted up until 1920, implying that mushroom picking remained a popular practice. Some interesting uses of famine plants—including the underground organs of wild plants—occurred in the 19th century, and the use of wild vegetables by rural communities was still in evidence into the 20th century [23–26]. Obviously wild plants were also used for human nutrition during the Irish famine (1840–1841) [39].

In Britain, foraging continued as a pastoral pursuit up to and throughout the Second World War. The Ministry of Agriculture and Fisheries published a popular guide to edible and poisonous fungi from 1910 (first edition) to 1945 (sixth edition) that was popular enough to require two further reprints in 1947 and 1950 [38]. During the war, urban children were evacuated to the countryside and many had their first experiences of brambling (collecting blackberries) and related pursuits [40,41]. At this time, rosehips were gathered to provide an extra source of vitamin C. However, during the 1950s and 1960s, with the onset of intensive agriculture, society's focus was on the new and old traditions slipped further away.

In the 1970s, a growing "back to nature" movement developed in the U.K. Authors such as Seymour [17], Mabey [24], and Duff [42] published literature on the use of wild plants. In the 1990s, recipes for foraged foods were popularised by British chefs [43]. Television programs such "Cook on the Wild Side" (1995 and 1997) and River Cottage's foraging segments (1999–2009) brought the experience of foraging to a wide national audience, and in 2004 Oxford Brooke University held their Symposium on "Wild Food: Hunters and Gatherers" [34].

Interest in foraging and wild foods in the British Isles has steadily grown since then. Meanwhile, within the last two decades, a large number of papers were published in other European countries to document their disappearing foraging traditions, summarised by Łuczaj and colleagues [3]. In some countries such as Poland, Belarus, Sweden, Estonia, Croatia, and Hungary, it was possible to trace the changes in foraging traditions due to numerous historical sources from the 19th century and the first half of the 20th century [44–50]. A host of ethnogastronomic publications also come from Spain, Italy, and Turkey [51–53].

Wild food and foraging have been researched and documented from a number of different perspectives. Many contributions fall within the research area "non-timber forest products" (NTFPS) [54]—despite the fact that wild foods come from a diverse range of habitats, not just from forests. This approach recognises the economic importance of wild

foods in local economies but often fails to acknowledge the social, cultural, psychological, and spiritual aspects of foraging. Others highlight the practice as one that supports factors such as psychological wellbeing and cultural heritage, alongside economic practices [55].

Foraging movements are even present today in city agglomerations throughout the world, and initial attempts to publish papers on these activities have been made, e.g., in Seattle (USA) [56,57], Berlin (Germany) [58,59], and London [60]. Moreover, there is a wide range of reasons behind urban gathering. It can be a part of established local culture [61,62], practiced by children looking for snacks [63], or caused by famine or war [64]. Urban studies have also been carried out on the toxicology of wild edible plants gathered in city areas (e.g., San Francisco) [65]. A new research area is also emerging on the recent popular trend of gathering wild foods among health-conscious people [3]. We have seen first publications concerning the motivation for gathering wild foods in Austria [18] and Spain [16]. Lee's thesis is the earliest attempt to study people foraging in Britain [5]. Recently a study of contemporary wild plant foraging in Czechia was also published [66].

A large, public source of information—both written and pictorial—that records the main dynamics of this trend is available online through social media sites, such as Facebook and Instagram (Figure 1), as well as via published books and websites. The original authors of this content are also some of the leaders of today's foraging movement. These are people writing books on wild edible plants, making TV programmes, and organising foraging and cookery workshops. As their popularity grows, the "culture" that they create becomes more celebrated, with particular plant uses and dishes becoming fashionable, e.g., amongst chefs in high-end restaurants.

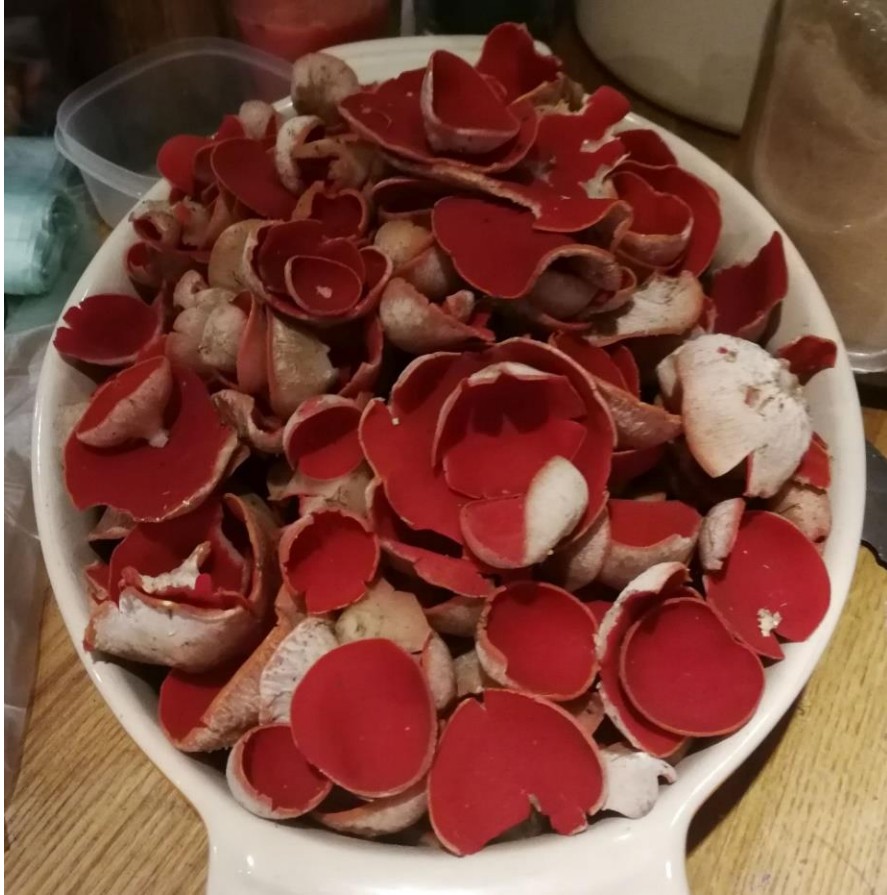

**Figure 1.** Elf cups (*Sarcoscypha* sp.) are edible fungi often shown on Instagram due to their vivid red colours. Although they have little taste, they are easily recognised and look beautiful in photos. Photo by Łukasz Łuczaj.

In the British Isles, the growing popularity in foraging for wild foods is driven by a number of factors including, but not limited to, a desire to spend time in and interact with nature, an increasingly "foodie" culture and the practices of fine dining restaurants which follow in the footsteps of pioneers such as Fearnley-Whittingstall [67], and a drive towards more localised food systems and sourcing more sustainable food [68]. The use of wild foods in restaurant settings is particularly prevalent in nature-based tourism destinations, where a growth in "food tourism" highlights the importance of more sustainable, locally sourced and produced food. In these cases, wild food is celebrated as the pinnacle of local and sustainable produce and adds a kind of gastronomic authenticity to the products on offer [69]. Wild food is taking a prominent role in conversations about food provenance, where a higher value is increasingly placed on food which is harvested or produced as close to the source as possible [70]. Thus, wild food is an important part of the "terroir" concept reinvented by Nordic Cuisine and the Slow Food movement in general [71,72] Examples such as this highlight the role that wild foods might take in rural (and urban) economic development, in connecting food with local place identities, and in contributing to alternative food systems that highlight local food products and the sustainable use of natural resources [73–76]. Wild foods are an increasing component of restaurant menus. It is not only haute-cuisine restaurants serving them [3] but also local restaurants that offer regional foods [69,70]. Unfortunately, we do not have publications directly recording foraged foods served in British and Irish restaurants.

Despite the drop in widespread popularity of foraging in the 19th and 20th centuries, a thriving community of people that both practice foraging and teach it to other people exists in the British Isles today. The growing popularity of foraging supports the growth of a professional wild community of instructors who run workshops teaching laypeople how to safely forage for wild foods, as well as how to prepare those foods in the kitchen for fresh and preserved use. Additionally, this professional community is home to a growing number of entrepreneurs working with wild foods in the delivery of wild food and drink products, and services. These include restaurants and "pop up" dining events dedicated to wild foods, drinks such as specialist gins and liqueurs, preserves, and the supply of fresh wild plants and fungi [15].

Many of these wild food professionals are members of the Association of Foragers (AoF), an organisation started in 2015 to promote professionalism and provide support to members. The idea originally arose from Andy Hamilton, John Rensten, Monica Wilde, and Mark Williams, early leaders teaching foraging in England and Scotland, who were joined by a further 21 founding members at a "meet-up" in Bristol in 2015 (including Łuczaj and Wilde). This became an unincorporated association that rapidly grew to around 130 members by 2020.

The instigation of a formal association was twofold. Firstly, members enjoyed the experience of community, knowledge exchange, communication, and fellowship. Secondly, foragers realised that access to public land for foraging was facing challenges from local government and landowners. Although collecting wild vegetables and fruits along paths and roadsides is generally non-problematic, some clashes arise, especially in areas managed by local government and conservation institutions (they are listed in detail in the Discussion part of the paper, Section 4.3). A formal association created a visible body with a public voice to debate some of the growing restrictions to foragers—such as denial of access to public land when wanting to forage or lead classes. Many foragers felt strongly that they had a higher purpose to reconnect people with nature, and issues of land rights over the commons have been hotly debated within the group at many meet-ups. Having created a formal organisation with formal "principles of practice" that members have to comply with, the association proactively leads on issues such as sustainable harvesting, land access, safety, conservation, and other matters that could easily become part of future byelaws and legislation [77].

As there have not yet been any papers attempting to give the overall characteristics of the foraging movement in the British Isles, the aim of this paper was to carry out the first

description of it. Our objective was to perform a detailed survey of the members of the Association of Foragers that could give us answers to the following research questions:

- Who are the leaders of the British foraging movement?
- What is the origin of their knowledge, their inspiration?
- What are the most frequently promoted wild foods? Are they traditional British foods, foreign traditions, or novel uses?
- Do foragers experience problems with gathering wild food?
- How do they assess the impacts of their gathering?

## 2. Materials and Methods

To accurately characterise the innovation and influence of modern-day foraging, we selected a cross section of people teaching foraging or organising small-scale gathering of wild foods. Large-scale producers of a few of the most commonly gathered wild products were not included. In the course of the study, a few methods were used: participant observation, in-depth qualitative interviews, and a structured questionnaire.

Łuczaj, the lead author of the paper, took part in three gatherings of the AoF (December 2015 in Bristol; 2018 in Dorset; 2020 in Tan-y-Cefn, Rhayader, Wales) and in some smaller informal meetings—e.g., a meeting of foragers in Kent in 2019, as well as a course organised by one of the foragers. The author also personally knows over 50 foragers from all parts of the British Isles and has had numerous conversations with them during the last 6 years. Moreover, he has been regularly visiting the UK since 1997 and has spent some time working or living in Norwich, Somerset, and Glasgow. He also organised foraging workshops in Norwich, Norfolk, in 2002 and 2003. Throughout this time, he has had ample opportunities to see how the foraging scene in Britain has been developing. The second author, Wilde, a Fellow of the Linnean Society, has been teaching foraging for 17 years, and has been a founding member and committee member of the Association of Foragers since its inception. She personally knows most of the association members and has attended all of its annual meetings. The final author, Townsend, has been foraging in Scotland for many years and teaching for the last 4. She is a member of the Association of Foragers as well as a social scientist currently researching foraging practices in Scotland.

In January 2019, the questionnaire (Appendix A) was distributed to the foragers gathered at the Annual Meeting in Ardkinglas, Scotland. Seventeen of the members responded in writing. Later, 3 interviews were gathered in a meeting in Kent, and 6 more at the Annual Meeting in Wales in February 2020 (Figure 2). Ten more questionnaires were gathered later from other AoF members who either forgot to fill them in during the above-mentioned meetings or could not attend, but are nevertheless active and early members of the association. The questions are itemised in Appendix A. Altogether, we included 36 respondents (including the second author of the paper).

Most foragers have basic to advanced skills in plant and fungi identification. We were not able to collect voucher specimens of the listed taxa, but the respondents often provided the botanical names of the plants. To avoid any doubt, only the genus name was used if the entry was unclear. Plant taxonomy follows the Plant List [78], fungi—Index Fungorum [79], and algae—AlgaeBase [80]. As there is no standardised list of scientific and authority names for all animals, the names followed those used in specialist publications in Britain for specific animal orders.

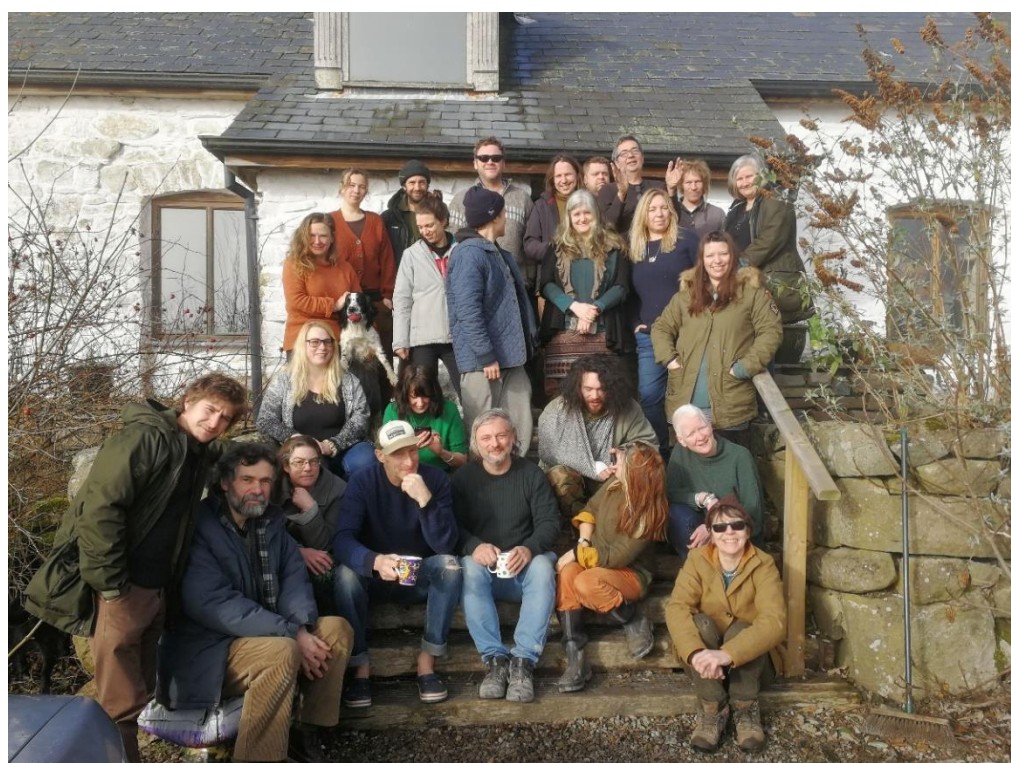

**Figure 2.** A group photo from the most recent Association of Foragers (AoF) meeting in Tan-y-Cefn, Rhayader, Powys, Wales, 3–7 Febuary 2020. Photograph by Łukasz Łuczaj.

## 3. Results

### 3.1. Social Characteristics

Both sexes are well represented among British leading foragers (19 males, 17 females). Eight of them have a BSc or MSc in biology, herbal medicine, or ecology. Some people declared degrees in other disciplines (e.g., physics, gastronomy, arts, anthropology, or psychology). Nine people declared they have not got any "related skills" and are entirely self-taught. The average age of the respondents was 49—median 48.5, minimum 29, maximum 74. They had on average 9 years of experience in teaching foraging.

The foragers run courses in various parts of the British Isles, but the southern part of England as well as Scotland are the most covered (Table 1).

**Table 1.** The regions where foraging courses are usually run and where the interviewees are based.

| Region or Country | Where They Are Based—Number of People, $n = 36$ | Where They Run Courses—Number of People, $n = 36$ |
|---|---|---|
| East of England | 3 | 3 |
| East Midlands | 1 | 1 |
| Republic of Ireland | 5 | 5 |
| London | 2 | 5 |
| North West England | 1 | 1 |
| Scilly Islands | 0 | 1 |
| Scotland | 8 | 10 |
| South East England | 5 | 11 |
| South West England | 11 | 10 |
| Wales | 1 | 3 |
| Yorkshire and the Humber | 2 | 2 |

### 3.2. Most Commonly Gathered and Taught Wild Foods

The plants used by foragers for their personal use (102 taxa or categories) and those taught in their courses (100 taxa or categories mentioned) were found to be nearly identical (Table 2). The Sørensen's similarity index was above 98%. The instructors usually draw immensely from their own everyday experience and use similar plants at home.

**Table 2.** Species most commonly used by foragers in their courses and in their private life.

| Scientific Name | English Name | Category | Used in Courses, *n* = 36 | Used Personally, *n* = 36 | Parts |
|---|---|---|---|---|---|
| *Allium ursinum* L. | Wild Garlic, ramsons | higher plant | 24 | 24 | ap, fl |
| *Sambucus nigra* L. | Elder | higher plant | 24 | 24 | fl, fr |
| *Urtica dioica* L. | Nettle | higher plant | 24 | 24 | ap, fr |
| *Heracleum sphondylium* L. | Common hogweed | higher plant | 13 | 13 | ap, fr |
| *Rubus* subgenus *Rubus* | Blackberry | higher plant | 12 | 13 | fr |
| *Beta vulgaris* L. | Sea beet | higher plant | 11 | 11 | ap |
| *Taraxacum* spp. | Dandelion | higher plant | 11 | 11 | ap |
| *Cantharellus* spp. | Chanterelle | mushroom | 11 | 11 | fu |
| *Allium triquetrum* L. | Three cornered garlic | higher plant | 10 | 10 | ap |
| *Palmaria palmata* (L.) F. Weber & D. Mohr | Dulse | algae | 10 | 10 | ap |
| *Crataegus* spp. | Hawthorn | higher plant | 9 | 9 | fr, ap |
| *Stellaria media* L. | Chickweed | higher plant | 9 | 9 | ap |
| *Rosa* spp. (mainly *R. canina* L.) | Rose | higher plant | 9 | 8 | fr, fl |
| *Alliaria petiolata* L. | Garlic mustard | higher plant | 7 | 7 | ap |
| Fungi | Mushrooms in general | mushroom | 7 | 7 | fu |
| *Malus* spp. | Wild apple | higher plant | 7 | 7 | fr |
| Algae | Algae in general | algae | 6 | 6 | ap |
| *Smyrnium olusatrum* L. | Alexanders | higher plant | 6 | 6 | ap |
| *Boletus edulis* | Bolete | mushroom | 6 | 4 | fu |
| *Galium aparine* L. | Cleavers | higher plant | 5 | 5 | ap |
| *Elaeagnus rhamnoides* (L.) A. Nelson | Sea buckthorn | higher plant | 5 | 5 | fr |
| *Hydnum repandum* | Hedgehog mushroom | mushroom | 5 | 5 | fu |
| *Laminaria digitata* (Hudson) J.V. Lamouroux and other Laminariales | Kelp | algae | 5 | 5 | ap |
| *Rumex acetosa* L. and *R. acetosella* L. | Sorrel | higher plant | 5 | 5 | ap |
| *Sonchus* spp., mainly *S. oleraceus* L. | Sow thistle | higher plant | 5 | 5 | ap |
| *Agaricus* spp. | Field mushroom | mushroom | 4 | 4 | fu |
| *Tripolium pannonicum* (Jacq.) Dobrocz. | Sea aster | higher plant | 4 | 4 | ap |
| *Himanthalia elongata* (L.) S.F. Gray | Thongweed | algae | 4 | 4 | ap |
| *Porphyra linearis* Greville and *P. umbilicalis* Kützing | Laver | algae | 4 | 4 | ap |
| *Prunus spinosa* L. | Blackthorn | higher plant | 4 | 4 | fr |

ap—aerial parts (aboveground parts in the case of land plants), fl—flowers, fr—fruits and seeds, rt—roots or other underground organs.

The foragers teach and use mainly wild vascular plants, with the addition of some sea algae, as well as fungi. Animal food is used rarely. The five most commonly foraged wild foods are wild vegetables: wild garlic *Allium ursinum* L., elder *Sambucus nigra* L., nettle *Urtica dioica* L., common hogweed *Heracleum sphondylium* L., and blackberry *Rubus* spp. The most commonly used fungi are chanterelle *Cantharellus* spp., bolete *Boletus edulis*, and hedgehog mushroom *Hydnum repandum.* The most commonly used algae are dulse *Palmaria palmata*, kelp *Laminaria* spp., and thongweed *Himanthalia elongata*.

The choice of food proposed by the instructors for consumption in the case of a catastrophe is similar to that mentioned above (Table 3). As many as 102 species or food types are used. More animals (20 taxa compared to 5) and plants with underground edible organs are mentioned (5 species compared to 1).

**Table 3.** Species that could be used in case of a sudden catastrophe/emergency to satisfy hunger—according to the respondents.

| Scientific Name | English Name | Frequency (*n* = 36) | |
| --- | --- | --- | --- |
| *Urtica dioica* L. | Nettle | 22 | ap |
| Phaeophyta and Rhodophyta | Algae in general | 14 | ap |
| Fungi | Fungi in general | 12 | fu |
| *Sambucus nigra* L. | Elder | 10 | fl, fr |
| *Taraxacum* spp. | Dandelion | 8 | ap |
| *Beta vulgaris* L. | Sea beet | 6 | ap |
| *Heracleum sphondylium* L. | Hogweed | 6 | ap |
| *Rubus* subgenus *Rubus* | Blackberry | 6 | fr |
| *Smyrnium olusatrum* L. | Alexanders | 5 | ap, fr |
| *Stellaria media* L. | Chickweed | 5 | ap |
| *Corylus avellana* L. | Hazel | 5 | fr |
| - | Fruits in general | 5 | fr |
| *Quercus* spp. | Oak | 5 | fr |
| *Arctium lappa* L. | Burdock | 4 | rt |
| Cervidae | Deer in general | 4 | mt |
| *Allium* spp. | Wild garlic in general | 4 | ap, fl |
| *Allium ursinum* L. | Wild garlic, ramsons | 4 | ap, fl |
| *Chenopodium album* L. | Goosefoot | 4 | ap |
| *Castanea sativa* L. | Chestnut | 4 | fr |
| *Elaeagnus rhamnoides* (L.) A. Nelson | Sea Buckthorn | 4 | fr |
| *Rosa* spp. | Wild rose | 4 | fr |

ap—aerial parts (above ground parts in the case of land plants), fl—flowers, fr—fruits and seeds, rt—roots or other underground organs.

Outside the plant, algae, and fungi kingdoms, some small sea animals are consumed on seashore courses (Figure 3). Insects and other land invertebrates were hardly mentioned.

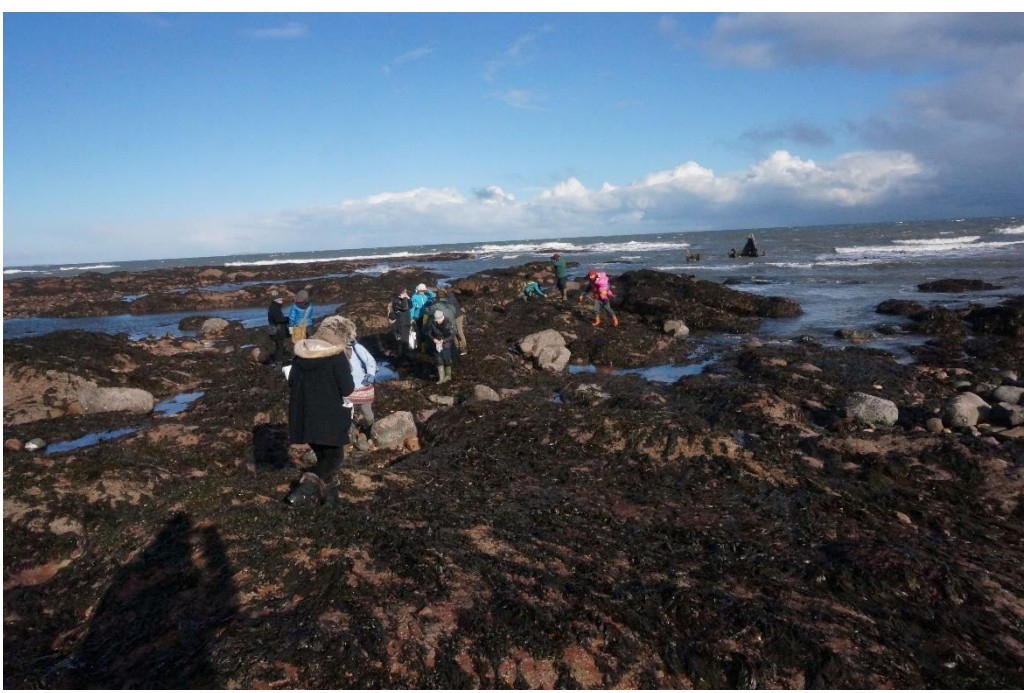

**Figure 3.** Seaweed workshops run by Monica Wilde on the East Scottish coast, February 2016. Photo by Łukasz Łuczaj.

### 3.3. Roadkill

Roadkill is consumed by only some of the foragers (22 out of 36), both by men and women. (Eight men and six women had never tried roadkill.) Men listed many more species (4.1) compared to women (1.7), although the difference was not statistically significant (Mann–Whitney $U$ test, $p = 0.25$). Pheasant (Figure 4), various species of deer (mainly roe deer), rabbit, hare, badger, and grey squirrel are the most frequently reported kinds of roadkill (Table 4). Altogether, 34 species were recorded, out of which there were 18 mammals (including one case of a dolphin killed by a boat), 15 birds, and 1 fish (dropped by a bird).

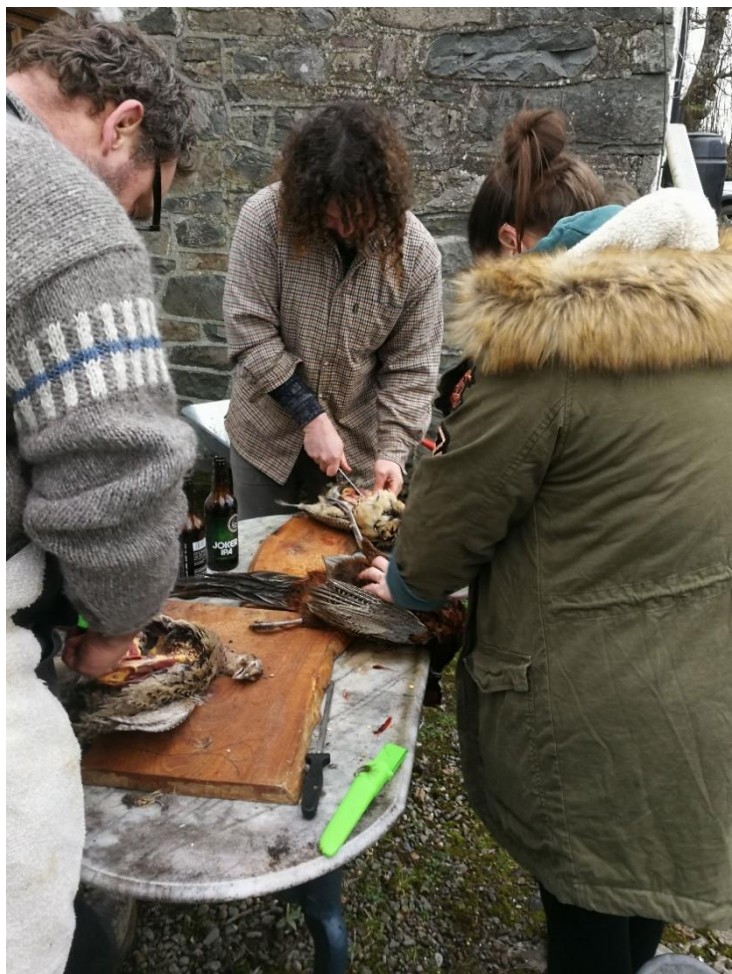

**Figure 4.** Preparing roadkill pheasants served during the AoF meeting, Wales, February 2020. Photo by Łukasz Łuczaj.

**Table 4.** Species of roadkill that foragers have eaten.

| Common Name | Scientific Name | Frequency ($n = 36$) |
| --- | --- | --- |
| Pheasant | *Phasianus colchicus* | 17 |
| Deer | Cervidae | 11 |
| Rabbit | *Oryctolagus cuniculus* | 9 |
| Roe deer | *Capreolus capreolus* | 7 |
| Grey squirrel | *Sciurus carolinensis* | 5 |
| Hare | *Lepus* sp. | 5 |
| Badger | *Meles meles* | 5 |
| Hedgehog | *Erinaceus europaeus* | 4 |
| Red deer | *Cervus elaphus* | 3 |
| Partridge | *Perdix perdix* | 3 |

**Table 4.** *Cont.*

| Common Name | Scientific Name | Frequency (*n* = 36) |
|---|---|---|
| Muntjac deer | *Muntiacus* sp. | 3 |
| Fox | *Vulpes vulpes* | 3 |
| Fallow deer | *Dama dama* | 3 |
| Sitka deer | *Odocoileus hemionus sitkensis* | 2 |
| Pigeon | Columbidae | 2 |
| Mallard ducks | *Anas platyrhynchos* | 2 |
| Brown hare | *Lepus europaeus* | 2 |
| Wood pigeon | *Columba palumbus* | 1 |
| Trout (from a regulated cormorant, it was on a road) | Salmoninae | 1 |
| Thrush | *Turdus philomelos* | 1 |
| Red grouse | *Lagopus lagopus scotica* | 1 |
| Otter | *Lutra lutra* | 1 |
| Mute swan | *Cygnus olor* | 1 |
| Moorhen | *Gallinula chloropus* | 1 |
| Mole | *Talpa europaea* | 1 |
| Manx shearwater | *Puffinus puffinus* | 1 |
| Lapwing | *Vanellus vanellus* | 1 |
| Kangaroo | *Macropus rufigriseus* | 1 |
| Irish hare | *Lepus timidus hibernicus* | 1 |
| Brent goose | *Brenta bernicla* | 1 |
| Dolphin (boat kill) | Delphinidae | 1 |
| Collared dove | *Streptopelia decaocto* | 1 |
| Cat | *Felis catus* | 1 |
| Bullfinch | *Pyrrhula pyrrhula* | 1 |
| Blackbird | *Turdus merula* | 1 |
| Red-legged partridge | *Alectoris rufa* | 1 |

*3.4. Poisonings and Indigestion*

Only 15 out of 36 foragers reported that indigestion or a minor poisoning problem was experienced first-hand during their courses. Altogether, there were 19 dangerous situations or indigestion recorded, plus 7 cases mentioned as "heard of" from other foragers (Table 5). Out of 18 taxa mentioned, half (9) were fungi and 8 were vascular plants. Each taxon was mentioned by only one respondent at a time, apart from seaweeds. Three respondents mentioned that seaweeds were a rare cause of indigestion among course participants. A few course organisers mentioned single situations of one person having an upset stomach without an apparent reason (Table 5). The problems can be mainly divided into two categories: minor indigestion (usually after eating fungi or seaweed dishes) and allergic reactions.

*3.5. Sources of Inspiration and Knowledge*

Foragers' own experimentation; knowledge from books, websites, social media, and films; and the knowledge of other experts constituted similarly important sources of knowledge (34.5% ± SD = 23.2, 34.3 ± 21.0, and 28.5 ± 19.4%, respectively). Knowledge acquired from traditional uses by locals in rural areas or people overseas seemed to play a minor role (9.7 ± 11.2%).

When asked which people had inspired them, the leading positions were held by Roger Philips, Richard Mabey, Miles Irving, Fergus Drennan, Monica Wilde, and Mark Williams (all of the former are British foragers), as well as the American forager Pascal Baudar. Among the varieties of media types, published books dominated almost exclusively, including many American authors.

Respondents listed few names as their direct personal teacher (only 17), and only Miles Irving was listed three times, with three other people mentioned twice. A further 47 names were mentioned as people who had inspired the foragers. Fergus Drennan was mentioned

11 times, followed by Roger Phillips (10 times), Monica Wilde (9), Mark Williams (9), Miles Irving (8), Richard Mabey (5), and Pascal Baudar (4).

**Table 5.** Cases of minor poisonings, allergies, or potential poisonings during foraging.

| Species | Syndrome |
| :---: | :---: |
| *Arum maculatum* L. | a person NEARLY ate the leaf confusing it with sorrel |
| *Clitocybe nebularis* (Batsch) P. Kumm. | minor indigestion |
| *Coprinus comatus* (O.F. Müll.) Pers. | minor indigestion |
| *Craterellus cornucopiodes* (L.) Pers. | minor indigestion |
| *Ficaria verna* Huds. | itchy throat |
| *Fuchsia* sp. | coughing after eating the fruits |
| *Galium aparine* L. | three course participants affected over 5 years |
| *Heracleum sphondylium* L. | body rash after touching it with lips on hot day |
| *Hydnum* sp. | breasts swollen up (allergic reaction) |
| *Impatiens glandulifera* Royle | scratchy throat |
| *Lactarius* sp. | minor indigestion |
| *Laetiporus sulphureus* (Bull.) Murrill | minor indigestion |
| *Lepista personata* (Fr.) Cooke *(L. saeva)* | minor indigestion |
| *Leucopaxillus giganteus* (Sowerby) Singer | minor indigestion |
| *Rumex acetosa* L. | dizzy, queasy |
| Seaweed | minor indigestion, three people over 5 years |
| Seaweed | minor indigestion |
| Seaweed | minor indigestion |
| *Sparassis crispa* (Wulfen) Fr. | only one in a 13 person group affected |
| *Umbilicus rupestris* (Salisb.) Dandy | a person suffering from kidney stones had kidney pains |
| *Urtica dioica* L. | tingling mouth from eating the seeds |

There was also a large amount of experimentation mentioned during the interviews conducted (i.e., trying different processing techniques, recipes, plants parts, or even new species). Very little is taken from indigenous populations and most AoF members have either no (or very limited) experience with living with indigenous groups, foraging, or cooking with them.

*3.6. The Law*

Only eight foragers have never had any problems gathering food or running courses. Most foragers have experienced minor problems, such as occasional individuals concerned about the sustainability of the harvest. These were often solved or explained on the spot. Cases where problems were encountered only concerned southern England, predominantly coastal parts of Kent (problems with collecting sea kale in a site of special scientific interest (SSSI)) and the New Forest in Hampshire (see Section 4.3 for a discussion of these disputes). People from Scotland and Ireland in particular emphasised that they did not experience problems with their activities (although by-laws in Phoenix Park, Dublin, state that foraging is not permitted—albeit not heavily enforced).

*3.7. Sustainability*

Most people were confident that their activity has little impact on the environment. Only a few cases of observed overharvesting in the U.K. were mentioned by eight people. Altogether, 14 taxa or group of taxa were mentioned. Four taxa were mentioned twice, these being elder fruits (*Sambucus nigra*), ceps (*Boletus edulis* and related taxa), sea kale (*Crambe maritima*), and chaga mushrooms (*Inonotus obliquus*). Other taxa mentioned were *Allium ursinum, Batis maritima*, "berries in towns" in general, *Crithmum maritimum, Juniperus communis, Mertensia maritima, Morchella* sp., *Myrrhis odorata*, and *Salicornia europaea*. Seaweeds (*Laminaria* spp.) are destroyed by mechanised harvesting rather than by individual harvesters. A third of the mentioned species are restricted to coastal ecosystems. In some cases, the concern was about the very existence of the species in certain locations (wild

vegetables by the sea), in others about the availability to other foragers or animals (shrubs with edible fruits and ceps).

## 4. Discussion

### 4.1. What Is Collected and What Could Be Collected

Among the most commonly used fruits, all of them have a place in the history of British cuisine. Moreover, a large part of vegetables, such as sea beet *Beta vulgaris*, *Urtica dioica*, *Smyrnium olusatrum*, *Rumex* spp., and *Chenopodium album*, were once consumed in the British Isles, although not all [21–26]. For example, hogweed *Heracleum sphondylium* is a commonly consumed plant but its use has no local tradition.

British foragers seem to be particularly reserved about using underground storage organs of plants. Such organs are the mainstay of hunter-gatherer societies and were commonly used in Europe by peasantry, especially in times of food shortages. In Britain, the underground starchy organs of *Potentilla anserina*, *Arctium* spp., Orchidaceae, *Crambe maritima*, *Eryngium maritimum*, and *Arum maculatum* were once consumed [23–26]. However, foraging courses usually present fruits and leaves as "wild food", avoiding underground parts, as under the Wildlife and Countryside Act 1981, digging out underground organs is illegal [81]. In the questionnaire, underground organs were only mentioned (very rarely) in the question about plants used during a catastrophe, e.g., one person mentioned *Arctium* and *Arum maculatum.* This is in contrast to foraging courses in other countries, e.g., Poland, where bulbs and roots are occasionally dug out by participants (unpublished data from Łukasz Łuczaj). It seems that British foragers do not treat wild plants as their main source of calories but rather a source of flavouring, vitamins, and various micronutrients—replacing conventional vegetables and mushrooms, adding interest to the plate, or simply mystically experiencing nature. This focus on above-ground parts of the commonest plants is also understandable as many foraging courses are run in urban or peri-urban spaces and the organisers want to minimise the impact of the courses on nature.

Another area where foragers are extending British food traditions is the use of seaweeds. Seaweeds were used as food, especially famine food in the British Isles [82], but now this tradition is being revived and enhanced by recipes from East Asia.

Fungi are frequently taught and eaten in foraging courses. Eating fungi in Britain was very little practiced in the past compared to mycophilous parts of Europe (the east and the south). For example, in one area of Poland, 76 species of mushrooms have been used as food, and in other parts of Poland, around 20 species are reported as commonly used in various local ethnographic papers (for an overview, see [83]). Here, foragers are definitely the leaders of change in the culinary practices of the British population that are interested in nature. Collecting fungi does cause controversy in some more densely populated parts of the British Isles. Some people perceive collecting the fruiting bodies of mushrooms as depleting nature and spoiling the countryside. Foragers are sometimes questioned by passers-by, verbally attacked, or even banned (often against the law) from collecting mushrooms (see Section 4.3 for further discussion). The main area of conflict about mushroom collecting is the New Forest. In contrast to this fear of collecting mushrooms, there is a growing general consensus among mycologists that collecting fruiting bodies of mushrooms does not really endanger their population and can even spread them. Instead, it is trampling the forest floor that can damage the mycelia [84].

Eating roadkill by foragers is an interesting issue as it is a widespread source of the meat used for their personal purposes. Up to the present, we have not found any scientific publications documenting the use of roadkill in societies. This is a new phenomenon associated with the development of roads and practiced, not from poverty, but as a part of the sustainable use of local resources. Arthur Boyt, from Davidstow, Cornwall, should be seen as precursor of roadkill eating in the British Isles. He has feasted on roadkill badgers for decades and gave many media interviews about it [85]. Hunting in the U.K. is popular, however, it requires funds and access to land, and thus it is mainly delivered as

an organised, paid-for activity through large estates and landowners. Therefore, it is not taken up by foragers under the 'food for free' premise.

Many aspects of using roadkill as food are unclear—e.g., assessing its microbiological safety, ownership of the dead animals, etc. That is why this is done on a small scale and not as part of workshops for outsiders. Scotland's Natural Larder (a Scottish Natural Heritage organisation) has made some training materials on handling game safely (http://scotlandsnaturallarder.co.uk/game/; accessed on 1 March 2021); however, there was political pressure not to publicly address the issue of roadkill, although this was discussed in conversation with Wilde (the second author).

The most commonly collected species are those already present in the British culinary tradition, such as pheasant, deer, hare, or rabbit. Some innovations included grey squirrels and some rarer birds. Eating of foxes recorded in three responses is interesting. Its meat was described by the interviewees as not particularly pleasant and needing long hours of cooking and roasting. By experimenting on themselves, the foragers are an avant-garde of European cuisine, in a different way than the top restaurants. The foragers often reach for untested items regarded by the public as inedible or toxic. By showing that they can be eaten, they broaden our understanding of the food resources available to humankind. Unfortunately, insects, recently branded as the food of the future, are almost completely neglected by foragers. They were mentioned only twice as "insects" and "insect larvae", without specifying the species (although we know from in-depth interviews that a few of the foragers use ants). This probably stems both from a culinary 'insectophobia' displayed by the British public and by the low availability of insects decimated by modern intense agriculture and pesticide use.

### 4.2. Making the Knowledge

Surprisingly, the foragers generally have had no contact or little contact with people gathering wild food in other parts of the world. Foragers are often environmentally friendly focused people who would like to decrease their carbon footprint. Many of them do not even travel by plane and mainly explore their own area, although they do tend to drive to other parts of Britain to run courses. A strong source of their knowledge is literature, and to some extent their own experimentation. The same literature sources and similar people are repeatedly mentioned as sources of inspiration. Adding to this is the fact that most active British professional foragers are members of the AoF and many get together and attend the AoF meetings. This exchange provides a large common body of knowledge and practice created by the foragers that has formed a new "culinary culture" of hybrid origin. It joins local, sometimes temporarily forgotten, British culinary and foraging customs with inspirations from books on edible plants and foragers' own experimentation. In Table 6, we compared the use of particular food categories with the traditional use in Britain, Europe, and Asia.

**Table 6.** Comparison of main kinds of plants and fungi foraged by British foragers and people in traditional cultures in different parts of Eurasia.

| | Foragers | Great Britain | Eastern Europe, e.g., Poland | Mediterranean (e.g., Italy, Croatia, Greece) | East Asia (China, Japan, Korea) |
|---|---|---|---|---|---|
| Fruits | yes | yes | yes | yes | yes |
| Leafy vegetables | many species | only a few species | only a few species | many species | many species |
| Fungi | yes | nearly none | yes | usually yes | yes |
| Seaweed | a few species | a few species in the past | no | no | yes |

Out of the seven most inspiring people for foragers, four were members of the association, including one, Irving, who wrote the book *The Forager Handbook* [86]. Two were the older British book writers Phillips [25] and Mabey [24]. The former is a member, but the

latter is not a member of the AoF. The fourth is an American forager and book writer [87]. The high popularity of Drennan is interesting as he has not published any books to date. However, he is very active on social media, experimenting with new, sometimes unusual recipes, which he eagerly shares. When asked about inspiration, the informants mentioned 51 items: 19 foragers mentioned Roger Phillips' books, particularly *Wild Food* [25], as well as his books about fungi; 11 mentioned Richard Mabey's *Food for Free* [24]; 7 Ray Mears' TV programmes and books; while 6 mentioned Miles Irving's *The Forager Handbook* and Ken Fern's *Plants for a Future* [19]. The above-mentioned books and TV programmes seem to be a canon shared by most of the foragers.

The foragers' own cultural prejudices may have an effect on the choice of plants. For example, bracken has a very bad reputation in Britain as a possible carcinogen [88]. Although the plant is widely eaten in China, Japan, and Korea, and the foragers know this, they generally do not use either bracken or other ferns, although there are records of the practice in old newspapers [89] and especially in Hampshire and Scotland [90]. Thus, the old British culinary practice, utilising multiple fruits, nettles, sorrel, bistort, and selected seaweeds, is enriched selectively. At the moment, it is the mushrooms and seaweed that fascinate as something new, valuable, and tasty, whereas some practices are still disregarded (eating ferns, eating underground organs).

What is worth mentioning is that common hogweed is very popular among foragers now, yet it is not a part of the traditional British cuisine. Its use, now largely forgotten, was once mainly restricted to Eastern Europe (Poland, Lithuania, Belarus) [49,50].

According to our observation, there is a large level of homogenisation of recipes used by the foragers that results from publishing their dishes on social media or through personal communication. For example, *Heracleum sphondylium* fruits are often used as a spice e.g., for gingerbread, and many foragers also make "wild vermouths", "wild curry", "wild kimchi", "wild sushi", "wild gomasio" (Figure 5), and jelly ear "jaffa cakes".

### 4.3. Problems with Negative Attitude towards Foraging and Worries about Sustainability

Serious issues for local foragers around the use of plants and fungi are illustrated by experiences in Kent, where Miles Irving was served a stop notice in 2016 on a Natural England site that he had commercially collected plants from since 2003 [91]; in popular forests like the New Forest, where the public bodies have tried to ban foraging; and in the city of Bristol, which tried to ban foraging in 2016. These cases are discussed in detail below.

### 4.3.1. The New Forest

The New Forest, covering 28,003 hectares [92], is 90% owned by the Crown, and is managed by Forestry England (previously known as the Forestry Commission) and the National Trust, who owns further parcels of common land. The New Forest has a history of antagonism to foraging. Brigitte Tee-Hillman, a New Forest resident, has picked and sold mushrooms since 1973, on the basis of common rights law going back to the 15th century. In 2002, she was arrested and charged with mushroom theft. She appealed, and over 4 years (and 32 court appearances) the case was thrown out, costing the taxpayer around GBP 1 million. She went on to win a civil suit against the Forestry Commission—who had to pay all the case costs—and a special licence to collect fungi by the Department for Environment Food and Rural Affairs. Lee [15] points out that due to the "judge's dismissal of the case, the precise position for commercial foragers remains uncertain". Nevertheless, by 2006, all commercial mushroom collecting was banned (with the exception of Mrs. Tee-Hillman) [93], but the New Forest Fungi Code, allowing a personal collection limit of 1.5 kg a day, continued [94]. This was subsequently, post-2016, considered a "loophole" that needed to be closed [95].

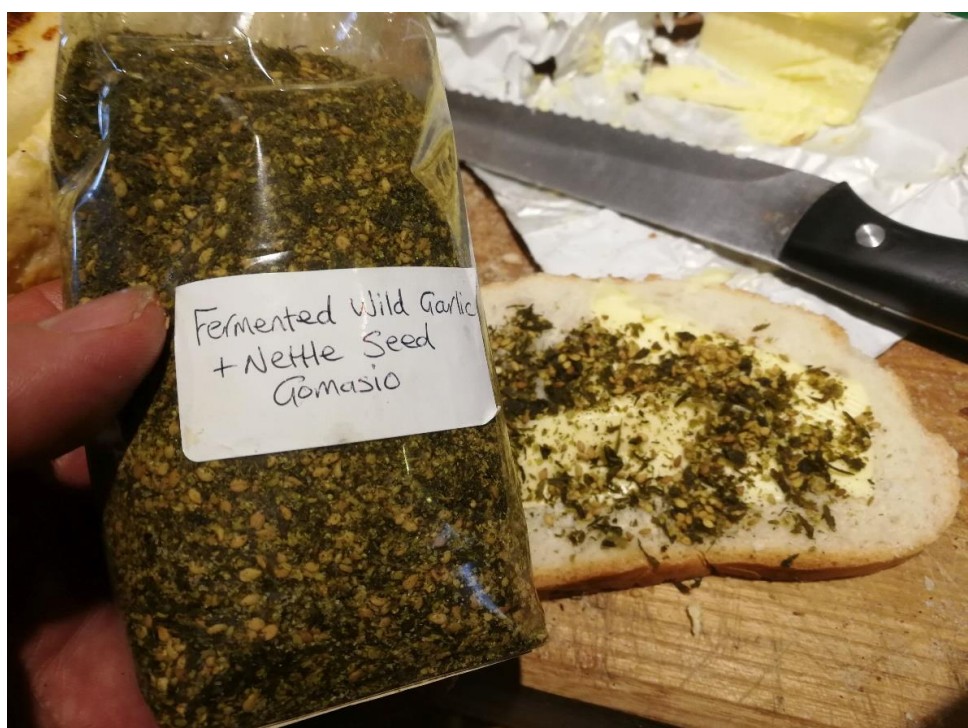

**Figure 5.** An example of "fusion" wild food used by British foragers—fermented wild garlic and nettle seed gomasio, served in the AoF meeting in Wales, 2020. Wild garlic (*Allium ursinum*) and nettles (*Urtica dioica*) are common native plants collected for generations, but in this case, they are used in a different context. The garlic is lactofermented (which is a completely forgotten technique in British cuisine). Nettle seeds are not the part of the plant that was traditionally used. The recipe itself is Japanese. Gomasio is a sesame salt; here, sesame was replaced by native ingredients. Photo by Łukasz Łuczaj.

In 2016, Forestry England decided, without any public consultation, to ban all fungi picking in the New Forest, allegedly due to a rise in illegal commercial collection of fungi by "gangs" noted by the Court of Verderers over 2014 and 2015 [96]. However, when challenged by members of the AoF, neither co-owner the National Trust—whose byelaws prohibit fungi picking [97]—nor Forestry England were able to provide any actual evidence of these gangs, nor any evidence to support their claim that some illegal pickers were making GBP 2500 per day [95]. The AoF was able to make some headway and foraging teachers can now apply for an annual licence to run mushroom identification forays while the general public are now politely "requested" not to pick fungi [98]. None of the public landowning bodies were able to confirm to AoF members that the 'ban' had a legal basis.

The legality of the restriction is a moot point and laws and byelaws often contradict each other.

- The Forestry Commission Byelaws 1982 used by Forestry England restricts the cutting or removing or plants but does not mention fungi [99].
- Clause (2b) in the National Trust Byelaws 1965 is more prohibitive and states that "No unauthorised person shall dig up or remove, cut, fell, pluck or injure any flowers, plants, fungi, moss, ferns, shrubs, trees or other vegetation growing on Trust Property or remove any seeds thereof or injure any grass or climb any tree" [97].
- Conversely, The Theft Act 1968 Section 4 "Property" (3) says "A person who picks mushrooms growing wild on any land, or who picks flowers, fruit or foliage from a plant growing wild on any land, does not (although not in possession of the land) steal what he picks, unless he does it for reward or for sale or other commercial purpose. For purposes of this subsection "mushroom" includes any fungus, and "plant" includes any shrub or tree." [100].

- The Criminal Damage Act 1971, s.10 (1) repeats that for purposes of the Act, property "does not include any mushrooms or the flowers, fruit or foliage of any plant growing wild on land (s. 10(1)(b)), so long as the plant is not uprooted or significantly damaged" and, additionally, it has no commercial purpose test [101].
- Both The Wildlife and Countryside Act 1981 and the definition of Sites of Special Scientific Interest (SSSI) require protected species to be specifically identified in their schedules [91].

However, there is clearly pressure from the residents of the New Forest to see bans enforced [102], and AoF members have reported being challenged by residents.

### 4.3.2. Epping Forest

The City of London Corporation (CLC) who owns Epping Forest covering 2500 hectares in Essex [103] started prosecuting people for picking fungi in 2014 and have been particularly active since then, signposting and policing the forest, and prosecuting fungi pickers. As with the New Forest, the rationale is often confusing. One reason given is that picking fungi threatens biodiversity and deprives animals such as deer of a food source. There is no scientific evidence that harvesting mushrooms does threaten biodiversity but arguably there are animals that do eat mushrooms. In a statement the CLC also posited that "Ancient trees also rely on fungus to protect their roots" and yet picking fungi does not disturb these fungal networks [104]. Moreover, of the 1500 fungi species in the forest [103], only a dozen species are commonly picked by the public.

### 4.3.3. Royal Parks

The Royal Parks banned foraging in 2010, stating in 2015 that "the number of people foraging fungi in the Royal Parks (is) thought to be on the rise" [105], and dramatically announced that incidents had gone up 600% in 2018 [106] to a total of 35 people who received police warnings [106]. Their rationale is that animals such as deer need chestnuts to eat during the winter months and that "picking mushrooms, the fruit of fungi underground, can hinder reproduction, limiting the ability of fungi to thrive". A Metropolitan police officer stated "We've noticed a steady rise in mushroom foraging over the years which could possibly be attributed to celebrity chefs' endorsement" [105] and "The parks are here for the public to enjoy—they are not anyone's personal larder" [107], demonstrating a predisposed intolerance of foragers.

### 4.3.4. Bristol

In February 2016, Bristol City Council proposed introducing new byelaws that would make foraging illegal [108]. However, this was spotted by local AoF members who campaigned to raise awareness of the plans. Subsequently, many opinions were publicly expressed, for example, "people have been picking the land for centuries and foragers know that damaging plants will mean that next year there won't be anything to forage". Objections were raised through the council's consultation process that had 813 respondents [109], of which 23% of the comments in Section 12 were in regard to the "Protection of structures and plants" with "many responses on berry picking and foraging". The council attributed this to "media coverage of the Byelaws" generating "a misconception around the freedom to pick berries in Parks and Green spaces." The end result was that the original unpopular proposed byelaw was discarded and revised [110].

Overall, in Southern England, although commercial foraging has been banned since 1968 under The Theft Act 1968, there also seems to be a growing trend to ban all personal picking rather than simply arrest unlicensed commercial pickers. Lee [15] notes that in England "an increasing amounts of land has come under the care of quasi-governmental organisations and their associated byelaws, this will de jure limit the land that is available for purposes of foraging."

Conversely in less-populated Scotland, foraging is not discouraged. Here, the landowning and governing bodies actively seek to educate on a responsible approach to harvesting

wild food. For example, in 2010, the government agency NatureScot [111] has produced a guide to the Scottish Wild Mushroom Code in both English and Polish. Furthermore, they started the initiative "Scotland's Natural Larder", which aims to "reconnect people with local and natural produce that has been harvested or hunted, encouraging Best Practice and responsible use of natural food resources." [112]. The Woodland Trust [113] has an online foraging guide, while Forestry and Land Scotland [114] gives advice on foraging on their website. This widely differing attitude is borne out by the experiences of AoF members.

It seems that in most situations where the above-mentioned concerns about the activity of foragers occurred, the resistance resulted from the lack of a living tradition of foraging. For example, in Poland, fungi collecting is very widespread and not regulated. People gather mushrooms even in peri-urban areas around Warsaw without conservation bodies reporting any damage [84]. In contrast, some other European countries, such as Spain, impose certain rules on mushroom foraging, e.g., harvesters pay a certain annual fee [115]. Moreover, most wild vegetables are collected freely in Southern Europe. For example, in Croatia, only the gathering of *Asparagus acutifolus* is associated with paying special license fees. In 2013, this payment was introduced for all people gathering the species but due to large opposition from the public, in 2018, it was restricted only to those selling the species commercially [116]. Thus, in trying to introduce regulations on foraging, local authorities tend to take into account its economic character. We argue that foraging of common species should be encouraged as a part of building food security and connecting people with nature, whereas commercial harvesting on a large scale should be closely looked at and, in some cases, perhaps even regulated.

Arguably, the only case that should lead to more detailed scientific assessment is foraging seashore plants. Coastal areas harbour many rare plants that do not grow inland. Thus, gathering them can have a negative impact on their population.

### 4.4. Food Security or Fashion

Recently, Grivins [55] introduced a classification of foragers. He divided them into rooted, lifestyle, subsistence, and commercial foragers. Amongst the foragers we studied, a mixture of all four styles exists, but the lifestyle foragers dominate. The concern that foraging is only a part of trendy lifestyle, of a middle-class trend to show off, diversify cuisine, or introduce new or forgotten flavours was expressed by Richard Mabey (the famous author of *Food for Free*) in his article in *The Guardian* [117], "Most telling maybe has been the change in the fortunes of that delectable seashore succulent, marsh samphire. In the sixties it was an arcane, liminal food, a 'poor man's asparagus'. In 1981 it was served at Charles and Diana's wedding breakfast, gathered fresh from the Crown's own marshes at Sandringham. Now (imported from Brittany) it's a widespread garnish for restaurant fish, an indigenous seaside holiday souvenir, sold by the bag to those who don't want to get their own legs mud-plastered, and along the north Norfolk coast, an occasional bar snack, lightly vinegared in bowls next to the crisps and pickled onions."

Within a few days, he got a strong response from Miles Irving, one of the pioneers of the current foraging movement in Britain (and currently a member of the Association of Foragers) [118] who believes that foraging can substantially contribute to British food security. Wild food is a low-calorie source of food, and Britain is a densely populated country. Even if Irving's claim is over-optimistic, wild foods may contribute to improving the diet of the population by enabling people to self-gather wild vegetables rich in micronutrients, vitamin C, and folic acid. We argue that there is a need for a broader discussion regarding how citizens can play an active role in designing more sustainable food systems and what are the policy/civil society mechanisms in Britain that allow for that to happen. At the moment, there seem to be two camps: those enthusiastically taking up foraging for personal interest and as a solution to a much broader problem of sourcing sustainable and unpolluted food, and those who see foraging as a threat to the nature of the British Isles, which is already seriously depleted by high population density, urban sprawl, and intense agriculture.

Furthermore, in times of humanitarian crisis—as proved during the siege of Sarajevo—wild plants have been crucial in helping to alleviate food shortages [64]. For this reason, an army project took place in the former Yugoslavia to explore the potential of wild food. Soldiers were trained to subsist on it along the coast, but the knowledge was later dispersed amongst the general public [119]. What is more, collecting wild foods may be a good way to re-introduce knowledge about basic common species of organisms common in the British Isles. Recently a book titled *The Lost Words* showed pictures of plants and animals which disappeared from the Oxford Junior Dictionary as they are no longer used by young people [120]. Many of the depicted plants are wild edible plants such as nettle, dandelion, or blackberry. The foraging movement—providing that foragers can control the potential for overharvesting of rare species—may thus be a way to counteract a trend that has been called the "devolution of traditional knowledge" [121].

## 5. Conclusions

Foragers in Britain are a relatively homogenous group in terms of the origin of their knowledge, which is usually a combination of learning from literature, self-experimentation, and learning from colleagues. A lot of recipes and ideas are circulating among them that quickly spread via modern media, which leads to homogenisation of the knowledge and creates a post-industrial foraging cuisine. This includes both local traditions and exotic influences concerning the choice of species and recipes. This culture is created rapidly mainly by the access to social media and personal exchanges between the foragers, partly through the AoF meetings. The exchange has grown rapidly since the establishment of the association.

The core of the species prepared and served as food during courses are common wild vegetables, fruits, mushrooms, and seaweeds. Only a few consume sea animals and roadkill. Poisoning cases are very rare, and all of these to date have simply been minor indigestion.

Most foragers are well integrated and understood by their local communities and they do not experience legal problems with collecting wild foods. It is only in a few areas of (predominantly) southern England that some members of local communities and some landowners are against foraging, imagining that it will deplete nature. This fear is not grounded as most foragers have adopted the "principles of practice" of the AoF in order to become members, and they teach and promulgate sustainable and responsible harvesting. Future research could explore the contested spaces that foragers inhabit in order to understand the debate around land access and sustainability from diverse perspectives—including those of foragers, land managers, conservationists, and laypeople. As the growth of foraging continues to rise, so too does its profile, and it will be necessary for different stakeholders to reach shared understandings in the years to come. Hopefully, the foraging movement in Great Britain and Ireland, integrated by the existence of the AoF, can manage to find a sustainable form of using nature for the benefit of local communities and individuals. The AoF can provide the function of an intermediary between all the interested stakeholders, i.e., amateur and commercial foragers, foraging teachers, landowners, and nature conservation institutions, encouraging those institutions to understand that foraging is widely practiced in many countries without damaging the environment and should be supported in a reasonable form, rather than banned.

**Author Contributions:** Conceptualisation, methodology, writing—original draft preparation, Ł.Ł. and M.W. (equal contribution); writing—review and editing, Ł.Ł., M.W. and L.T. All authors have read and agreed to the published version of the manuscript.

**Funding:** This research received no external funding.

**Institutional Review Board Statement:** Ethical approval was not required by the lead authors institutions.

**Informed Consent Statement:** Informed consent was obtained from all subjects involved in the study.

**Data Availability Statement:** The data matrix was deposited in the Repository of Rzeszów University https://repozytorium.ur.edu.pl/handle/item/6014, accessed on 25 February 2021. It includes answers to 12 questions (no. 5–11, 13, 15–19)—those which do not contain personal data that could identify them.

**Acknowledgments:** Many thanks to all the Association of Foragers (https://foragers-association.org, accessed on 25 February 2021) members who took part in the study.

**Conflicts of Interest:** The authors declare no conflict of interest apart from the fact that they are foragers actively involved in making foraging a wider practice in the society.

## Appendix A. Questionnaire

In addition to personal information questions (about age and gender), the following questions were asked:

1. Where are you based?
2. Where did you spend your childhood?
3. How many years have you worked as a foraging instructor?
4. What are your qualifications?
5. Please list below up to ten of the most frequent or abundant ingredients of wild food you use for your own nutrition
6. What are the most commonly foraged items used on your courses (up to ten)?
7. What wild products would you reach for in times of economic catastrophe?
8. What species of roadkill have you used?
9. Have you ever been poisoned by wild food yourself? With what?
10. Have you ever experienced people getting indigestion, allergies or poisonings on your own courses? With what species and in what circumstances?
11. Did you have any masters who taught you directly?
12. Have you been inspired by any foragers?
13. Were there any books, films or websites that influenced you?
14. In which regions do you run your courses?
15. Could you estimate the percentage of your knowledge based on:

    (a) your own experiments . . . . . . . . . . %
    (b) literature/media . . . . . . . . . . ..%
    (c) knowledge from other experts you know . . . . . . . . . . . . %
    (d) knowledge from contacts with locals from your area or when travelling . . . . . . . . %

16. Are there any things that, in your opinion, should change in the law concerning collecting wild food?
17. Are there any species that are being overexploited by foragers? If yes, which?
18. Have you ever had an opportunity to learn from tribal people or people living in rural areas? Where? What did you learn?
19. Have you ever had problems from other people or institutions with collecting plants from areas open to the public (lane edges, parks, state forests, coast etc.)? Can you describe them?

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
