# Peer review of "The Ethnobiology of Contemporary British Foragers: Foods They Teach, Their Sources of Inspiration and Impact"

_sustainability, doi:10.3390/su13063478_

Round 1
Reviewer 1 Report
The manuscript “The ethnobiology of contemporary British foragers: foods they teach, their sources of inspiration and impact” is a well-structured and well-prepared study. The new fashion of foraging is growing worldwide, so it is crucial to understand the teacher’s drivers and their sources of inspiration, and the impact they make.
There some small things that could be improved:
Line 33-34 I would generalize – in times of crises, e.g. war or famine
Line 34-34 Reference needed
Would be great to get more on new food fashion – use of wild foods in the high-rank restaurant.
Line 156-157 – no need to stress first and second, readers can count, just name the authors.
What about the governance arrangements towards foraging, is it free access, or permitted somehow? Please write a sentence before paragraph line 159.
I like the research questions, but please as well define the aim and objectives of the study.
Table 1 no need for that table if in the results there is no connection to regions of Britain.
In the questionnaire you asked where did you spend your childhood, but if they use wild foods in the childhood and experience the foraging practices?
Line 238 Animal food is used rarely, what do you mean? Hunting is a very popular activity in Britain.
Line 260 Vp – vegetative parts – maybe use term Aerial parts?
Maybe to use some illustrations, no figures at all in the manuscript?
Road killed animals- very interesting, they have eaten fox, cat and hedgehog, did the respondents specify how they prepared it or any feedback on taste? The hedgehog is used in gypsies’ cultures but never heard about eating a cat.
In the conclusions please add about the knowledge transmission, their sources of inspiration, and impact.
Author Response
Review 1
The manuscript “The ethnobiology of contemporary British foragers: foods they teach, their sources of inspiration and impact” is a well-structured and well-prepared study. The new fashion of foraging is growing worldwide, so it is crucial to understand the teacher’s drivers and their sources of inspiration, and the impact they make.
There some small things that could be improved:
Line 33-34 I would generalize – in times of crises, e.g. war or famine
>changed
Line 34-34 Reference needed
>we added the reference Łuczaj, Pieroni et al. here which quotes a list of famine food studies
Would be great to get more on new food fashion – use of wild foods in the high-rank restaurant.
>it was already discussed in the second half of the introduction, we reinforced it by adding two sentences.
Line 156-157 – no need to stress first and second, readers can count, just name the authors.
>changed
What about the governance arrangements towards foraging, is it free access, or permitted somehow? Please write a sentence before paragraph line 15
>These issues are discussed in detail in Discussion. We added a reference sending the reader there.
I like the research questions, but please as well define the aim and objectives of the study.
>We reformulated this part.
Table 1 no need for that table if in the results there is no connection to regions of Britain.
>We decided to leave this table. It provides some interesting info e.g. that most active foragers are missing from Midlands and Northern Ireland.
In the questionnaire you asked where did you spend your childhood, but if they use wild foods in the childhood and experience the foraging practices?
>Unfortunately we did not have this question but many of them declared such experiences in the non-structured parts of interviews. We however did not quantify it.
Line 238 Animal food is used rarely, what do you mean? Hunting is a very popular activity in Britain.
>We studied foraging – e.g. collecting food and not killing larger animals. But as a rule this is not practiced in foraging workshops
Line 260 Vp – vegetative parts – maybe use term Aerial parts
>yes, we agree, we changed it
Maybe to use some illustrations, no figures at all in the manuscript?
>We added a few photos now
Road killed animals- very interesting, they have eaten fox, cat and hedgehog, did the respondents specify how they prepared it or any feedback on taste? The hedgehog is used in gypsies’ cultures but never heard about eating a cat.
>they did not specify it apart from mentioning long processing. We added a section on foraging in the discussion.
In the conclusions please add about the knowledge transmission, their sources of inspiration, and impact.
The knowledge transmission and sources of inspiration is already mentioned in conclusions:
“Foragers in Britain are a relatively homogenous group in terms of the origin of their knowledge which is usually a combination of learning from literature, self-experimentation and learning from colleagues. A lot of recipes and ideas are circulating among them and quickly spread via modern media.”
But we added a sentence. We already wrote about the impact but we reinforced it by adding another sentence.
g the dangers of foraging and it impact
Reviewer 2 Report
The article looks at the increasing popularity of foraging in the UK. It investigates what species at the centre of modern foraging and the relationship between foraging and nature conservation or forestry managers. In so doing, it sheds light on this cultural phenomenon, its forms and impact.
Overall, the article reads well and is well structured. However, considering the focus of the review, it would be important the authors will reinforce the discussion about the link between these practices and the topic of sustainability. E.g. the discussion about potential clashes between foraging and conservation can be expanded both in the introduction and the discussion. Moreover, the article focuses on the UK. It appears, however, in tab 3.2 the reference to EIRE. Is it just a mistake and the authors wanted to indicate Northern Ireland instead - which is missing by the way in the tab - or rather what happens to North of Ireland? and can the authors tell more about what is the actual relationship between EIRE and the UK in the case of foraging?
Author Response
Review 2
The article looks at the increasing popularity of foraging in the UK. It investigates what species at the centre of modern foraging and the relationship between foraging and nature conservation or forestry managers. In so doing, it sheds light on this cultural phenomenon, its forms and impact.
Overall, the article reads well and is well structured. However, considering the focus of the review, it would be important the authors will reinforce the discussion about the link between these practices and the topic of sustainability. E.g. the discussion about potential clashes between foraging and conservation can be expanded both in the introduction and the discussion.
>We devoted a large part of the discussion on this clash and we did not want it to dominate the paper. But we added more stuff on sustainability. Probably a separate paper is needed to full discuss detailed nuances of these clashes. We believe that the overview we gave is enough to encourage other researchers on this topic.
Moreover, the article focuses on the UK. It appears, however, in tab 3.2 the reference to EIRE. Is it just a mistake and the authors wanted to indicate Northern Ireland instead - which is missing by the way in the tab - or rather what happens to North of Ireland? and can the authors tell more about what is the actual relationship between EIRE and the UK in the case of foraging?
>Throughout the text we specify the British Isles which geographically includes the whole of Ireland. Unfortunately we could not find active foragers from northern part of the island in our meetings. That is why we used Table 1 to illustrate the interesting geographical distribution of foragers focused mainly of southern England, Scotland and Republic of Ireland, with very few in Wales, N Ireland and the north/central England.
Reviewer 3 Report
General comments on overall background
Suggest the article is framed within the broader food system sustainability debates or at least within UK food system discussions. Why the renewed attention to foraging? What is the importance of the article? How can we ensure that foraging is lawfully undertaken and that guidelines are provided?
Abstract
Suggest ensuring that geographic reference to the UK is made upfront. e.g. "Foraging in the UK is an increasingly popular activity" as the next statement regarding the "general public" may not be applicable to other geographies.
Also, best to specify if "foraging" is for leisure, for home consumption or for sale in local markets.
Missing some background info regarding the reasons behind increased interest in foraging.
Line 12 - "Association of foragers". This information could be removed and provided later. What is more important is to explain that these foragers hold courses for the general public. Why? To raise awareness?
Line 14 "Mainly leafy vegetables, fruits, fungi and seaweeds are used during courses they run (altogether 102 species 15 were mentioned as frequently used). There were 34 species of roadkill animals listed, usually only 16 for their private use. Suggest rephrasing.
Introduction
General comment. The first paragraph would benefit from additional references and examples. Suggest Bharucha, Z.; Pretty, J. The roles and values of wild foods in agricultural systems. Philos. Trans. R. Soc.
B Biol. Sci. 2010, 365, 2913–2926. [CrossRef] [PubMed]
General comment: it might be useful to explain to readers what constitutes wild foraging e.g. herbs, fruits, plants (which parts?), mushrooms, insects/animals?
General comment: It would be helpful and interesting for readers to see a graphic representation of how knowledge of wild edibles has changed over time. Linked to major historical events.
Additional ref on the traditional and contemporary use/revival of wild edibles, Borelli et al. 2020. https://doi.org/10.3390/plants9101299
I also wonder how much attention is devoted in the UK to overharvesting of wild edibles, particularly in view of the resurging interest in this practice. Has any research been carried out or do sustainable guidelines for wild foraging exist? Do foragers of wild mushrooms, particularly Boletus (a very lucrative business in other parts of Europe) need a special permit? In Italy, for example, wild mushroom collection is regulated and collectors must use wicker baskets to ensure spore dispersal.
(lines 34-36)
"Producing high calorie crops to feed large populations using modern 34
farming techniques is efficient but it involves high consumption of fossil fuels and uses pesticides and herbicides that are harmful to the environment". Needs references, and some additional data.
(lines 61-62) Suggest rewording as the sentence is not gramatically correct. "Although the inhabitants of the British Isles are usually pictured as a prime example of mycophobia, the collection of edible wild mushrooms was locally practiced".
(lines 65-66) ... implying public popularity for the collection of edible fungi. Suggest changing to "Implying that mushroom picking remained a popular practice".
- Materials and methods
Lines 237-242: suggest providing common names next to the Latin, although they are provided in the table below.
Lines 299-303: Linked to this, it would be interesting to get a sense of the main reasons (by percentage) that push people to attend these lectures/lessons.
- Discussion
Line 346: Not sure I understand. Are the authors implying that foragers are reserved about owning up to the use of roots and tubers of certain plants because of the negative connotations linked to the use of these parts (e.g. food for the poor?) or because there are legal implications linked to digging up the roots? If so, consider explaining further.
Lines: 357-362 - Linked to suggestion above in Materials and methods about knowing earlier in the text about reasons for joining these foraging groups.
Table 6 - Is it possible to quantify "Many species"? There must be literature out there. Just to get a sense of the order of magnitude: hundreds, thousands?
Line 542-543 Suggest the distinction of the different types of foragers is also introduced earlier in the text, either in the intro or in the materials and methods.
Conclusions
Suggest providing suggestions of possible best practices/set of actions that can promote the use of wild edibles and help their conservation and sustainable use in Britain.

Author Response
Reviewer 3
General comments on overall background
Suggest the article is framed within the broader food system sustainability debates or at least within UK food system discussions. Why the renewed attention to foraging?
>We added 10 references in the introduction and several sentences.
What is the importance of the article? How can we ensure that foraging is lawfully undertaken and that guidelines are provided?
>The main message of the paper is that at the moment in UK the issue of sustainability of foraging in the UK is over-emphasized, we listed several examples of this in the discussion.
Abstract
Suggest ensuring that geographic reference to the UK is made upfront. e.g. "Foraging in the UK is an increasingly popular activity" as the next statement regarding the "general public" may not be applicable to other geographies.
> We moved ‘the British Isles’ to the first sentence. We did not want to repeat it due to limits of abstracts length
Also, best to specify if "foraging" is for leisure, for home consumption or for sale in local markets.
>this is really difficult to distinguish clearly, it is often for all 3 reasons
Missing some background info regarding the reasons behind increased interest in foraging.
>We cannot expand the abstract due to word count limits
Line 12 - "Association of foragers". This information could be removed and provided later. What is more important is to explain that these foragers hold courses for the general public. Why? To raise awareness?
>thanks for this comments, we changed ‘foragers’ to ‘foraging teachers’. We cannot expand the abstract more due to word count limits.
Line 14 "Mainly leafy vegetables, fruits, fungi and seaweeds are used during courses they run (altogether 102 species 15 were mentioned as frequently used). There were 34 species of roadkill animals listed, usually only 16 for their private use. Suggest rephrasing.
> We do not understand what is the problem with these sentences. The word count limit in the abstract is very short that is why we listed stuff in this monotonous way.
Introduction
General comment. The first paragraph would benefit from additional references and examples. Suggest Bharucha, Z.; Pretty, J. The roles and values of wild foods in agricultural systems. Philos. Trans. R. Soc.
B Biol. Sci. 2010, 365, 2913–2926. [CrossRef] [PubMed]
> We added this reference now. We do not want to add more references as the paper already has around 120
General comment: it might be useful to explain to readers what constitutes wild foraging e.g. herbs, fruits, plants (which parts?), mushrooms, insects/animals?
General comment: It would be helpful and interesting for readers to see a graphic representation of how knowledge of wild edibles has changed over time. Linked to major historical events.
> We do not have enough data for the UK to make more detailed graphs for the history of forgaging, this would require more reference and our paper is well over 100 rererences anyway.
Additional ref on the traditional and contemporary use/revival of wild edibles, Borelli et al. 2020. https://doi.org/10.3390/plants9101299
>This is a very interesting new reference, we added it, as well as Borelli et al. Local solutions for sustainable food systems: The contribution of orphan crops and wild edible species. Agronomy, 10(2), p.231.
I also wonder how much attention is devoted in the UK to overharvesting of wild edibles, particularly in view of the resurging interest in this practice. Has any research been carried out or do sustainable guidelines for wild foraging exist? Do foragers of wild mushrooms, particularly Boletus (a very lucrative business in other parts of Europe) need a special permit? In Italy, for example, wild mushroom collection is regulated and collectors must use wicker baskets to ensure spore dispersal.
>We devoted a large part of the discussion to explain the controversies in the UK. There are some local controversies in the south of England but they are not supported by any research. Actually, surprisingly, there is very limited evidence on the effect of mushroom foraging on the mushroom population anywhere. We quoted what is available. We do not have refs from Italy but we quoted the Spanish example about paying fees. In Britain no fees are needed but rights of collection depend much on the ownership of the land.
(lines 34-36)
"Producing high calorie crops to feed large populations using modern farming techniques is efficient but it involves high consumption of fossil fuels and uses pesticides and herbicides that are harmful to the environment". Needs references, and some additional data.
> we added some references. We cannot discuss this in detail though as this is not the topic of the paper.
(lines 61-62) Suggest rewording as the sentence is not gramatically correct. "Although the inhabitants of the British Isles are usually pictured as a prime example of mycophobia, the collection of edible wild mushrooms was locally practiced".
>we modified the sentence
(lines 65-66) ... implying public popularity for the collection of edible fungi. Suggest changing to "Implying that mushroom picking remained a popular practice".
> corrected
- Materials and methods
Lines 237-242: suggest providing common names next to the Latin, although they are provided in the table below.
>corrected
Lines 299-303: Linked to this, it would be interesting to get a sense of the main reasons (by percentage) that push people to attend these lectures/lessons.
>unfortunately we did not study these. These are not lectures – the question was about inspiring personalities within the foraging movement
- Discussion
Line 346: Not sure I understand. Are the authors implying that foragers are reserved about owning up to the use of roots and tubers of certain plants because of the negative connotations linked to the use of these parts (e.g. food for the poor?) or because there are legal implications linked to digging up the roots? If so, consider explaining further.
>The latter. We rephrased another sentence in the paragraph to make it clear: However, foraging courses usually present fruits and leaves as ‘wild food’, avoiding underground parts, as under the Wildlife and Countryside Act 1981 digging out underground organs is illegal.
Lines: 357-362 - Linked to suggestion above in Materials and methods about knowing earlier in the text about reasons for joining these foraging groups.
>We did not study the motivations. The motivations are often mixed and just the motovations could be a topic of another paper (we actually quote two papers, from Austria and Spain, about the motivations but wanted to avoid going into details).
Table 6 - Is it possible to quantify "Many species"? There must be literature out there. Just to get a sense of the order of magnitu de: hundreds, thousands?
> this is a general table in the discussion. Many means like 10-100 but it depends on the size of the group we study. I think it is self-explanatory for everyone and means “more than a few”
Line 542-543 Suggest the distinction of the different types of foragers is also introduced earlier in the text, either in the intro or in the materials and methods.
> We do not think the place matters, many foragers are mixed-type anyway.
Conclusions
Suggest providing suggestions of possible best practices/set of actions that can promote the use of wild edibles and help their conservation and sustainable use in Britain.
We wrote in the conclusions: “This fear is not grounded as most foragers have adopted the ‘Principles of Practice’ of the AoF in order to become members, and they teach and promulgate sustainable and responsible harvesting.” At the moment we think a bigger problem is over-emphasizing the dangers of foraging and it impact
Round 2
Reviewer 1 Report
Some small issues:
Figure 1 Time when the photo was taken and location.
Figure 2 Do you have at least the oral agreement of all the people to be published? and who did the photo?
Line 150 after Unfortunately should be a comma
Line 438 after Unfortunately should be a comma
Line 699 no need of dot after AoF
Author Response
Some small issues:
Figure 1 Time when the photo was taken and location.
>added
Figure 2 Do you have at least the oral agreement of all the people to be published? and who did the photo?
>The different versions of this photo circulate widely in the net and there was general consensus it is public. The photo is taken from my camera.
Line 150 after Unfortunately should be a comma
>corrected
Line 438 after Unfortunately should be a comma
>corrected
Line 699 no need of dot after AoF
>corrected
Reviewer 3 Report
While there was some effort to clarify the paper's main aim, the problem statement and the results of the research are still insufficiently framed. It is suggested that the final statement (below) in the most recent version provided is further developed and also worked into the abstract. What role can the AoF play going forward?
"Hopefully, the foraging movement in Great Britain and Ireland, integrated by the existence of the AoF can manage to find a sustainable form of using nature for the benefit of local communities and individuals. The AoF can provide the function of an intermediary between all the interested stakeholders".
Suggest rewording the abstract to something similar below (157 words at present). Problem statement and recommendations for future integration of AoF in sustainable food system dialogues would greatly benefit the discussion.
"Foraging in the British Isles is an increasingly popular activity for both personal consumption and for commercial purposes. While legislation and guidelines exist regulating the sustainable collection of wild edibles, the founding principles of the British foraging movement are not well documented. For this research, 36 of the most active foraging instructors of the Association of Foragers were interviewed to understand their background, species collected, sources of knowledge, and problems faced during collection. Altogether, 102 species of leafy vegetables, fruits, fungi and seaweeds were mentioned as frequently used, while 34 species of roadkills were listed, mostly for personal consumption. Instructors reported learning from wild food guide books, other foragers or personal experience. Frequent contacts among foraging groups has led to the standardisation of knowledge and practices among British foragers. Contrary to expectations, foragers rarely reported clashes with nature conservation or forestry managers. The authors argue that knowledge and guidelines developed by the AoF are sustainable and could be integrated..."
The other issues were more or less addressed.
Regarding overall word count: although important for context, the historical background could be shortened to provide space for what is perhaps the most original part of the paper: how citizens can drive and play an active role in designing more sustainable food systems and what are the policy/civil society mechanisms in Britain that allow for that to happen.
Author Response
Reviewer 3
While there was some effort to clarify the paper's main aim, the problem statement and the results of the research are still insufficiently framed.
>We do not understand in which way could we frame the problem statement better. In the results section we reported the results of our questionnaire. Please be specific if the change is really needed.
It is suggested that the final statement (below) in the most recent version provided is further developed and also worked into the abstract. What role can the AoF play going forward?
"Hopefully, the foraging movement in Great Britain and Ireland, integrated by the existence of the AoF can manage to find a sustainable form of using nature for the benefit of local communities and individuals. The AoF can provide the function of an intermediary between all the interested stakeholders".
>This role was already briefly summerised in the fragment you mentioned but we developed the paragraph to provide some ideas for the reader.
Suggest rewording the abstract to something similar below (157 words at present). Problem statement and recommendations for future integration of AoF in sustainable food system dialogues would greatly benefit the discussion.
"Foraging in the British Isles is an increasingly popular activity for both personal consumption and for commercial purposes. While legislation and guidelines exist regulating the sustainable collection of wild edibles, the founding principles of the British foraging movement are not well documented. For this research, 36 of the most active foraging instructors of the Association of Foragers were interviewed to understand their background, species collected, sources of knowledge, and problems faced during collection. Altogether, 102 species of leafy vegetables, fruits, fungi and seaweeds were mentioned as frequently used, while 34 species of roadkills were listed, mostly for personal consumption. Instructors reported learning from wild food guide books, other foragers or personal experience. Frequent contacts among foraging groups has led to the standardisation of knowledge and practices among British foragers. Contrary to expectations, foragers rarely reported clashes with nature conservation or forestry managers. The authors argue that knowledge and guidelines developed by the AoF are sustainable and could be integrated..."
>we followed your advice and modified the abstract, please check in the text if you are satisfied.
The other issues were more or less addressed.
Regarding overall word count: although important for context, the historical background could be shortened to provide space for what is perhaps the most original part of the paper: how citizens can drive and play an active role in designing more sustainable food systems and what are the policy/civil society mechanisms in Britain that allow for that to happen.
>We added a paragraph about it in the last section of the Discussion. We however, do not want to remove/shorten the historical background.